# Multi-locus phylogenetic network analysis of *Ampelomyces* mycoparasites isolated from diverse powdery mildews in Australia and the generation of two *de novo* genome assemblies

Lauren Goldspink[1], Alexandros G. Sotiropoulos[1], Alexander Idnurm[2], Gavin J. Ash[1], John D. W. Dearnaley[1], Morwenna Boddington[1], Aftab Ahmad[1], Márk Z. Németh[3,4], Alexandra Pintye[3,4], Markus Gorfer[5], Hyeon-Dong Shin[6], Gábor M. Kovács[3,4], Niloofar Vaghefi[7], Levente Kiss[1]*

1 Centre for Crop Health, University of Southern Queensland, Toowoomba, Queensland, Australia, 2 School of BioSciences, Faculty of Science, University of Melbourne, Australia, 3 Plant Protection Institute, HUN-REN Centre for Agricultural Research, Budapest, Hungary, 4 Department of Plant Anatomy, Institute of Biology, Eötvös Loránd University, Budapest, Hungary, 5 AIT Austrian Institute of Technology GmbH, Bioresources, Tulln, Austria, 6 Division of Environmental Science & Ecological Engineering, Korea University, Seoul, Korea, 7 School of Agriculture, Food and Ecosystem Sciences, Faculty of Science, University of Melbourne, Melbourne, Australia

* Levente.Kiss@unisq.edu.au

## Abstract

The interactions between powdery mildews (*Ascomycota*, *Erysiphaceae*), obligate biotrophic pathogens of many plants, and pycnidial fungi belonging to the genus *Ampelomyces*, are classic examples of specific mycoparasitic relationships. These interactions are common and finely tuned tritrophic relationships amongst host plants, powdery mildews, and *Ampelomyces* mycoparasites wherever these organisms co-occur in the field. Selected *Ampelomyces* strains have already been developed as biocontrol agents of powdery mildew infections of some crops. In Australia, their study has received little attention so far. Only a single *Ampelomyces* strain, included in a whole-genome sequencing (WGS) project, was known from this continent. Here, we report the isolation of 20 more *Ampelomyces* strains from eight powdery mildew species in Australia. Multi-locus phylogenetic network analyses of all the 21 Australian *Ampelomyces* strains carried out in combination with 32 reference strains from overseas revealed that the Australian strains belonged to four molecular taxonomic units (MOTUs). All those MOTUs were delimited earlier based on *Ampelomyces* strains isolated in Europe, North America, and elsewhere. Based on the phylogenetic analyses, two Australian strains belonging to different MOTUs were selected for WGS. Long-read (MinION) and short-read (Illumina) technologies were used to provide genome assemblies with high completeness. Both assemblies have a bipartite structure, i.e., consisted of AT-rich, gene-sparse regions interspersed with GC-balanced, gene-rich regions. These new high-quality assemblies and evidence-based

of the Creative Commons Attribution License, which permits unrestricted use, distribution, and reproduction in any medium, provided the original author and source are credited.

**Data availability statement:** All newly isolated fungal strains were deposited in a public culture collection, BRIP - accession numbers in Table 1. All newly determined DNA sequences were deposited in NCBI GenBank - accession numbers in Tables 1 & 2. All newly generated genome assemblies were deposited in NCBI GenBank - accession numbers in Table 3.

**Funding:** This study was supported by Discovery Project DP210103869 of the Australian Research Council and the University of Southern Queensland, Australia. There was no additional external funding received for this study.

**Competing interests:** The authors have declared that no competing interests exist.

annotations are important resources for future analyses of mycoparasitic interactions to disentangle molecular mechanisms underlying mycoparasitism, possible new biocontrol applications, and naturally occurring tritrophic relationships.

## Introduction

Fungi that attack and thrive on other fungi are commonly found in diverse environments. One of the well-known forms of aggressive interactions between fungal strains and species is mycoparasitism, which takes place when a fungus, the mycoparasite, feeds and develops its colonies on or inside another living fungus, the mycohost, and damages it through specific structural or other adaptations to this lifestyle [1–4].

The interactions between powdery mildews (*Helotiales*, *Erysiphaceae*), common obligate biotrophic pathogens of many plants [5–7], and pycnidial fungi belonging to the genus *Ampelomyces* (*Pleosporales*, *Phaeosphaeriaceae*) are classic examples of widespread mycoparasitic relationships in the field [8–10]. As powdery mildews can themselves be considered parasites of their host plants [11], *Ampelomyces* strains are also called hyperparasites, i.e., parasites of other parasites [8,9]. The finely tuned feeding interactions between host plants, powdery mildews and their *Ampelomyces* mycoparasites are well-known examples of specific tritrophic relationships in food webs [9,10,12–14]. These mycoparasites can be isolated from powdery mildew colonies and subcultured on agar media [15–21]. Selected strains of *Ampelomyces* mycoparasites were developed as commercial biological control agents of economically important powdery mildews that infect grapes and some vegetables [8,22,23] although biocontrol of powdery mildews with *Ampelomyces* strains applied as biological control agents was occasionally reported as poor or inconsistent [22].

The taxonomy of the genus *Ampelomyces* is still unresolved although molecular phylogenies based on the Internal Transcribed Spacer (ITS) region of the nuclear ribosomal DNA (nrDNA) and actin gene (*act*) fragments have identified distinct lineages within the genus [10,16–18,20,21]. The lineages were sometimes considered as molecular taxonomic units (MOTUs) [24]. These results indicate that the genus includes more than one species, not just *Ampelomyces quisqualis*, the type species; and all of those are mycoparasites of powdery mildews. The formal recognition of the MOTUs as other *Ampelomyces* species depends on the taxonomic treatment of over 40 taxa that are still validly described in the older mycological literature [8]. Until this is done, the use of *Ampelomyces* spp. is recommended when referring to phylogenetically diverse strains within the genus.

The asexual fruiting bodies, i.e., pycnidia, of *Ampelomyces* are commonly observed in field samples of diverse powdery mildew colonies, typically inside the cells of the powdery mildew conidiophores [8–10,16–21,25]. The asexual life cycle of *Ampelomyces* is favoured by humid conditions when mucilaginous matrices inside pycnidia take up water, swell, and conidia are released from powdery mildew colonies by the rupture of the pycnidial walls. Splash-dispersed conidia then germinate and the emerging *Ampelomyces* hyphae penetrate new powdery mildew hyphae on

the host plant surfaces. Following penetration, *Ampelomyces* hyphae continue to grow inside the powdery mildew mycelium, from cell to cell, and consume the powdery mildew cytoplasm. Finally, new pycnidia develop mostly inside powdery mildew conidiophores [9,25]. Hyphae of *Ampelomyces* can also be carried long distances inside parasitized powdery mildew conidia that are dispersed by air currents. When landed on powdery-mildew infected plants, these mycoparasitic hyphae may grow out of the airborne powdery mildew conidia and penetrate new mycohost colonies [8–10,26]. Both splash-dispersed and airborne *Ampelomyces* inocula can contribute to the spread of these mycoparasites to diverse powdery mildew species that are actively growing and infecting diverse host plants in the environment [9,10].

The extent of recombination amongst genetically diverse *Ampelomyces* strains is poorly understood. A fruiting body described as the sexual morph of *A. quisqualis* was reported once based on a field sample collected in Italy [27]. There are no other reports of the production of the sexual morph of *Ampelomyces* in the field or in culture. Nonetheless, population genetics analyses of hundreds of *Ampelomyces* strains performed with microsatellite or simple sequence repeat (SSR) markers revealed footprints of genetic recombination both within strains isolated from the same mycohost species, and those coming from diverse species of powdery mildew [19]. Similar signals of recombination have been detected in a number of other fungi that were previously thought to be asexual [28–36].

The above-mentioned study based on SSR markers [19] had also revealed genetic differentiation of *Ampelomyces* populations that are parasitising powdery mildews in spring versus the summer/autumn period in Europe. This was explained by temporal isolation of the respective populations rather than strict mycohost specialisations [10,19]. The possible mycohost specialisation of distinct *Ampelomyces* strains was studied in a number of laboratory and field experiments. Cross-inoculation tests indicated that *Ampelomyces* strains isolated from diverse powdery mildew species were able to parasitize other powdery mildew species tested as potential mycohosts [16,37–41]. This was also demonstrated in field experiments when potted cucumber and tobacco plants infected with *Podosphaera xanthii* and *Golovinomyces orontii*, respectively, were exposed to the attack of *Ampelomyces* strains that parasitised *P. leucotricha* on apple trees in that environment [10]. However, Falk et al. [39] observed a significantly higher rate of mycoparasitism in the original mycohost species of some strains compared to another powdery mildew species tested. Angeli et al. [42] reported that some, but not all, strains included in their experiments performed better in the original mycohosts than in other powdery mildews tested.

As in most ascomycetes, the hyphae and the conidia of *Ampelomyces* spp. are haploid. The chromosome numbers in *Ampelomyces* spp. have not been explored. The highest quality assembly and annotation of an *Ampelomyces* genome was published by Huth et al. [24] based on DNA long-read sequencing and total RNA sequencing (RNAseq) of a strain isolated from the powdery mildew species *G. bolayi* infecting *Cestrum parqui* in Queensland, Australia. The strain was identified as *A. quisqualis* and deposited as a live culture at the Queensland Plant Pathology Herbarium (BRIP) under the accession number BRIP 72107. Analyses of this genome revealed its bipartite structure with gene-rich, GC-balanced regions interspersed by long or short stretches of AT-rich, gene-sparse regions. This bipartite structure was also revealed in many plant pathogenic fungi and it is hypothesised to arise when duplicated DNA, such as transposons, undergo C to T transitions by the process of repeat-induced point mutation [43]. The substantial proportion of repetitive and AT-rich regions has been proposed to result in the 'two-speed' evolution of these genomes, where genes located close to the AT-rich regions have higher rates of evolution [44]. Based on these findings, Huth et al. [24] hypothesised that *Ampelomyces* mycoparasites may have evolved from plant pathogenic fungi.

The only other genome available for *Ampelomyces* mycoparasites is of strain designated HMLAC 05119 that was isolated from an undetermined powdery mildew infecting *Youngia japonica* in China [45]. Huth et al. [24] demonstrated that HMLAC 05119 is not conspecific with BRIP 72107 because the two strains belong to different MOTUs. However, HMLAC 05119 is also available under the binomial *A. quisqualis* in the NCBI GenBank database due to the yet unresolved taxonomy of the genus *Ampelomyces*. A near-chromosome level assembly was not generated for either HMLAC 05119 or BRIP 72107.

Most *Ampelomyces* strains included in different studies were isolated from diverse powdery mildew species infecting their host plants in Europe [10,12,15,18,19,37,42,46], Asia [16,17,20,21], and North America [39,47]. The highest number

of strains and field samples, over 600, was included in a population genetics study carried out by Pintye et al. [19]. To date, BRIP 72107 is the only *Ampelomyces* strain reported from Australia [24]. A few decades ago, *Ampelomyces* myco-parasites were observed in diverse powdery mildews in Australia with light microscopy [48], but to our knowledge the only strains that were isolated in this country are those included in the current work. A commercial *Ampelomyces* product was tested in Australia a few decades ago [49] without isolating the mycoparasites from the field.

To reveal the genetic diversity of *Ampelomyces* mycoparasites in powdery mildews in Australia, and select genetically different strains for further whole-genome sequencing (WGS) projects, the objectives of this study were to (i) isolate *Ampelomyces* strains from diverse powdery mildew and host plant species in Australia; (ii) perform a multi-locus analysis to reveal their phylogenetic relationships; and (iii) produce high-quality genome assemblies for Australian strains that belong to different MOTUs.

## Materials and methods

### Sample collections, isolations, and subculturing of *Ampelomyces* strains

Leaves and stems of different plant species infected with powdery mildew were collected *ad hoc* in southern Queensland, Australia, from 2017 to 2023. Powdery mildew colonies were examined under a stereomicroscope and a compound micro-scope for the presence of intracellular pycnidia characteristic of *Ampelomyces* in powdery mildew conidiophores. When found (Fig 1), pycnidia were removed from the powdery mildew colony one by one with sterile hand-made glass needles and each placed in a 6 cm diameter plate with saccharose-free Czapek-Dox medium supplemented with 2% malt extract (MCzDA) and 0.5% chloramphenicol [37]. Plates were incubated at 22°C in the dark and checked every 2–3 days for the emergence of small, slow-growing colonies that were tentatively identified as *Ampelomyces* colonies. Those emerging colonies were transferred to new plates as soon as they started to grow on the media. Pure cultures were maintained on MCzDA without chloramphenicol in an incubator at 22°C in dark and subcultured every 6–8 weeks on new plates.

Strains were deposited to the Plant Pathology Herbarium, Department of Primary Industries, Queensland, and named with their designated herbarium abbreviation (BRIP) and a number (Table 1).

### Identification of the mycohost powdery mildews

Powdery mildew species that were the mycohosts of the *Ampelomyces* strains isolated in this study were identified based on morphology, host plants and nrDNA ITS sequencing based on a nested PCR protocol as described by Kiss et al. [50]. The ITS sequences of six mycohost species, i.e., *Pseudoidium hortensiae*, *Arthrocladiella mougeotii*, *Podosphaera plantaginis*, *Golovinomyces bolayi*, *Erysiphe australiana* and *Podosphaera xanthii*, were identical to the ITS sequences determined in other powdery mildew specimens from the same host plants and deposited earlier in NCBI GenBank by Kiss et al. [50]. Based on morphology, the powdery mildew on *Parsonsia straminea*, the mycohost of BRIP 76210 and BRIP 76211; and the powdery mildew on *Salvia* sp., the mycohost of BRIP 76212, were identified as two distinct *Golovinomyces* species. Sequencing of the ITS region of these two powdery mildews with the nested PCR protocol [50] resulted in chromatograms that were uninformative due to poor quality. Therefore, the species identities of these two *Golovinomyces* specimens could not be determined.

### Mycoparasitic tests

The mycoparasitic activity of *Ampelomyces* strains, i.e., their penetration into powdery mildew hyphae and growth and development inside the hyphae and conidiophores, including the production of their intracellular pycnidia (Fig 1), were studied in laboratory, glasshouse and field experiments with different methods [10,16,22,40,41]. In this study, four *Ampelomyces* strains that sporulated well on MCzDA, i.e., BRIP 72097, BRIP 72100, BRIP 72107 and BRIP 72110, were selected for mycoparasitic tests. Mungbean (*Vigna radiata*) plants cv. Jade-AU infected with *Erysiphe vignae*, a powdery mildew

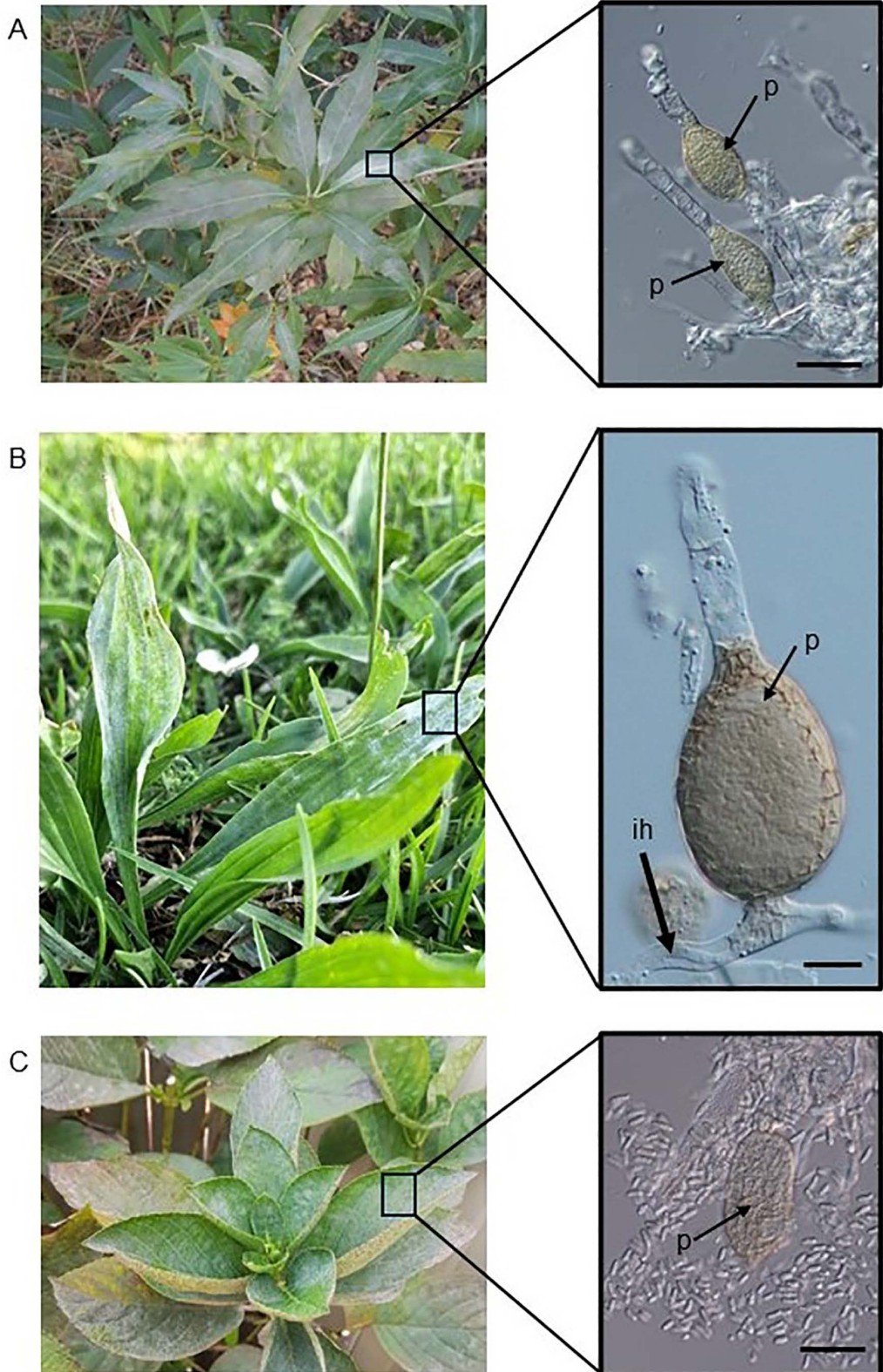

**Fig 1. Sources of the three Australian *Ampelomyces* strains, BRIP 72107, 72102, and 72097, with sequenced genomes.** A, *Cestrum parqui* infected with *Golovinomyces bolayi* at the collection site of *Ampelomyces* strain BRIP 72107. Inset: a part of the *G. bolayi* mycelium with pycnidia (p) in

the powdery mildew conidiophores. Bar = 30 μm. B, Ribwort plantain (*Plantago lanceolata*) infected with *Podosphaera plantaginis* at the collection site of *Ampelomyces* strain BRIP 72102. Inset: a conidiophore of *P. plantaginis* with a pycnidium (p) of *Ampelomyces* inside the foot cell. A fragment of an intracellular hypha (ih) of *Ampelomyces* is also visible inside the powdery mildew hypha. Bar = 10 μm. C, Bigleaf hydrangea or hortensia (*Hydrangea macrophylla*) infected with *Pseudoidium hortensiae* at the collection site of *Ampelomyces* strain BRIP 72097. Inset: An intracellular pycnidium (p) of *Ampelomyces* removed from the powdery mildew mycelium and surrounded by conidia released from it. Bar = 25 μm.

**Table 1. *Ampelomyces* spp. strains isolated from diverse powdery mildew species infecting different host plant species in Australia.**

| Strain designation | Host plant species | Host powdery mildew species | Place of isolation | Date of isolation | GenBank accessions | | |
|---|---|---|---|---|---|---|---|
| | | | | | ITS | *act1* | *eukNR* |
| BRIP 66222 | *Hydrangea macrophylla* | *Pseudoidium hortensiae* | Rangeville, Qld | 20-3-2017 | PQ813616 | PQ838553 | PQ838594 |
| BRIP 72097 | *Hydrangea macrophylla* | *Pseudoidium hortensiae* | Rangeville, Qld | 20-3-2017 | PQ813617 | PQ838554 | PQ838595 |
| BRIP 72098 | *Hydrangea macrophylla* | *Pseudoidium hortensiae* | Rangeville, Qld | 18-12-2017 | PQ813623 | PQ838561 | PQ838602 |
| BRIP 72099 | *Hydrangea macrophylla* | *Pseudoidium hortensiae* | Rangeville, Qld | 18-12-2017 | PQ813624 | PQ838562 | PQ838603 |
| BRIP 72100 | *Lycium barbarum* | *Arthrocladiella mougeotii* | Killarney, Qld | 13-10-2017 | PQ813626 | PQ838564 | PQ838605 |
| BRIP 72101 | *Plantago lanceolata* | *Podosphaera plantaginis* | Kearneys Spring, Qld | 18-12-2017 | PQ813633 | PQ838571 | PQ838612 |
| BRIP 72102 | *Plantago lanceolata* | *Podosphaera plantaginis* | Kearneys Spring, Qld | 18-12-2017 | PQ813634 | PQ838572 | PQ838613 |
| BRIP 72103 | *Plantago lanceolata* | *Podosphaera plantaginis* | Kearneys Spring, Qld | 18-12-2017 | PQ813635 | PQ838573 | PQ838614 |
| BRIP 72104 | *Cestrum parqui* | *Golovinomyces bolayi* | Newtown, Qld | 15-04-2019 | PQ813618 | PQ838555 | PQ838596 |
| BRIP 72105 | *Cestrum parqui* | *Golovinomyces bolayi* | Newtown, Qld | 15-04-2019 | PQ813619 | PQ838556 | PQ838597 |
| BRIP 72107 | *Cestrum parqui* | *Golovinomyces bolayi* | Newtown, Qld | 15-04-2019 | MZ054399 | PQ838557 | PQ838598 |
| BRIP 72108 | *Cestrum parqui* | *Golovinomyces bolayi* | Newtown, Qld | 15-04-2019 | PQ813620 | PQ838558 | PQ838599 |
| BRIP 72109 | *Cestrum parqui* | *Golovinomyces bolayi* | Newtown, Qld | 15-04-2019 | PQ813621 | PQ838559 | PQ838600 |
| BRIP 72110 | *Lagerstroemia indica* | *Erysiphe australiana* | Middle Ridge, Qld | 20-12-2018 | PQ813625 | PQ838563 | PQ838604 |
| BRIP 72111 | *Cestrum parqui* | *Golovinomyces bolayi* | Newtown, Qld | 22-10-2019 | PQ813622 | PQ838560 | PQ838601 |
| BRIP 72966 | *Vigna radiata* cv. Jade-AU | *Podosphaera xanthii* | near Pampas, Qld | 27-4-2021 | PQ813627 | PQ838565 | PQ838606 |
| BRIP 72967 | *Vigna radiata* cv. Jade-AU | *Podosphaera xanthii* | near Pampas, Qld | 27-4-2021 | PQ813628 | PQ838566 | PQ838607 |
| BRIP 72968 | *Vigna radiata* cv. Jade-AU | *Podosphaera xanthii* | near Pampas, Qld | 27-4-2021 | PQ813629 | PQ838567 | PQ838608 |
| BRIP 76210 | *Parsonsia straminea* | *Golovinomyces* sp.* | Irongate, Qld | 21-10-2022 | PQ813630 | PQ838568 | PQ838609 |
| BRIP 76211 | *Parsonsia straminea* | *Golovinomyces* sp.* | Irongate, Qld | 21-10-2022 | PQ813631 | PQ838569 | PQ838610 |
| BRIP 76212 | *Salvia* sp. | *Golovinomyces* sp.* | Preston, Qld | 3-11-2022 | PQ813632 | PQ838570 | PQ838611 |

*Based on morphology, the *Golovinomyces* species on *P. straminea* was different from the *Golovinomyces* species on *Salvia* sp.

fungus maintained in the laboratory [51], were used in these tests. Plantlets were grown from seeds in pots in an experimental glasshouse, in isolation in Bugdorm® cages as described previously [51]. When the first true, unifoliate leaves developed, plantlets were removed from pots, their roots rinsed with water, and placed each in a 50 mL Falcon tube filled with tap water and kept in a rack. On the same day, leaves were inoculated with *E. vignae* using powdery mildew-infected potted mungbean plants kept in Bugdorm® cages as described earlier [51]. Following inoculations, plantlets in Falcon tubes were kept in Bugdorm® cages in the glasshouse for 4–6 days, until their leaves were fully covered with sporulating powdery mildew mycelia; then, taken to the laboratory and sprayed with a conidial suspension of one of the four selected *Ampelomyces* strains using 5 mL plastic spray bottles. Conidial suspensions were produced by pipetting 3–4 mL water purified through reverse osmosis (RO), and autoclaved before use, onto 2–3 weeks old and sporulating *Ampelomyces* colonies in 6 cm diameter plates with MCzDA; then, rubbing their surfaces with a fine, sterile artist's brush to release as many conidia as possible from pycnidia. Suspensions were pipetted into 10 mL Falcon tubes from plates and their concentrations adjusted to approximately $10^6$ conidia/mL by dilution, after counting the spores with a Neubauer haemocytometer. Mungbean plantlets sprayed each with 3 mL *Ampelomyces* conidial suspensions, until runoff, were placed in transparent

plastic ziplock bags previously humidified by spraying 5 mL sterile RO water inside the bags; the bags were then kept for 7 days at 21-23°C and 16-hour daily illumination in a plant growth cabinet manufactured by Steridium Pty Ltd, Queensland, Australia (Fig 2A). Plantlets sprayed with 3 mL sterile RO water and kept in bags similar to the treated ones served as uninoculated controls. Conidial suspensions of each of the four selected *Ampelomyces* strains were sprayed each on two plantlets, i.e., four unifoliate leaves. Two plantlets served as uninoculated controls.

Seven days post inoculations (dpi) plantlets were taken out from bags and leaves were first examined under a dissecting microscope for the presence of *Ampelomyces* pycnidia in the conidiophores of *E. vignae*. When found (Fig 2B), a few pycnidia were removed one by one with glass needles and placed each in a separate plate with MCzDA supplemented with 0.5% chloramphenicol to re-isolate the mycoparasites and, thus, fulfill Koch's postulates [52]. Parts of the powdery mildew mycelium were then removed from each leaf with pieces of clear cellotape for further examination. Cellotape pieces were placed on microscope slides in droplets of 80% lactic acid. Slides were examined under a compound microscope to observe the fine details of mycoparasitism in the case of all four selected *Ampelomyces* strains. The experiment was carried out twice.

### DNA extractions, PCR amplifications and Sanger sequencing

Total genomic DNA was extracted from approximately 80–100 mg fresh weight mycelial samples taken from 3–4 weeks old colonies of each of the *Ampelomyces* strains. Samples were placed each in a 1.5 mL Eppendorf tube, freeze-dried, then ground to fine powder with two steel beads, 3 mm diameter. Grinding was done with a FastPrep-24 (MP Biomedicals, Australia) at 6.5 m/s for 30 s. The next steps were done using a DNeasy Plant Mini Kit (Qiagen, Australia) according to the manufacturer's instructions, except for the final step where DNA was eluted in 10 mM filter-sterilized Tris–HCl (pH 8.5).

Three loci were amplified from the genomic DNA and sequenced: the nrDNA ITS region; a fragment of the *act* gene; and a fragment of the nitrate reductase gene (*eukNR*). All three loci were amplified in 25 µL reactions with NEB HotStart 2× (New England Biolabs, Canada) using primers at a final concentration of 200 nM. Amplicons were submitted to Macrogen (Seoul, South Korea) for Sanger sequencing with the PCR primers. Sequences were deposited in GenBank (see accession details in Tables 1 and 2).

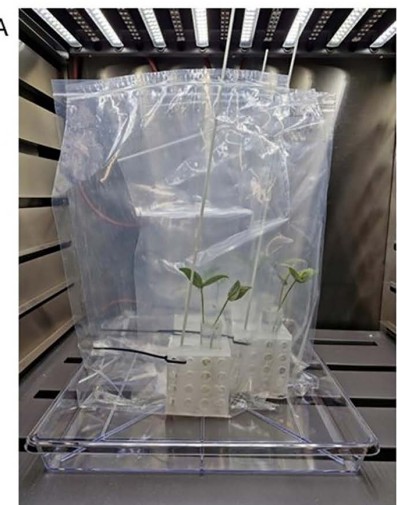 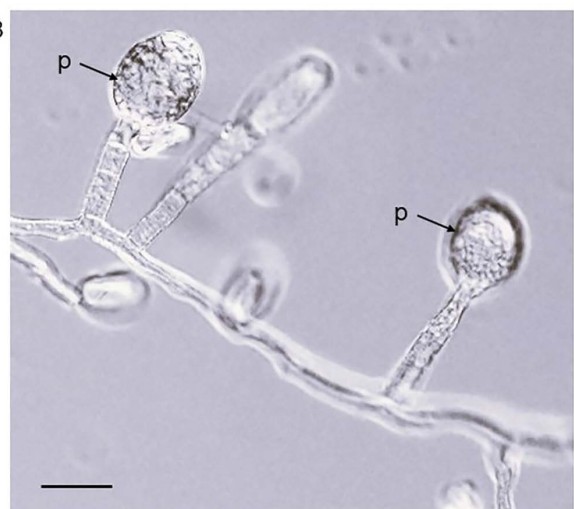

**Fig 2. A mycoparasitic test using mungbean (*Vigna radiata*) cv. Jade-AU plantlets with their first true leaves infected with the powdery mildew fungus *Erysiphe vignae* and inoculated with *Ampelomyces* strain BRIP 72097.** A, Each powdery mildew-infected plantlet was placed with its roots in water in a 50 mL Falcon tube kept in a rack inside a transparent ziplock bag. B, Pycnidia (p) of *Ampelomyces* developed in the conidiophores of *E. vignae* during the mycoparasitic test. Bar = 25 µm.

**Table 2. *Ampelomyces* spp. strains isolated outside Australia and included in this work as references. Source information was provided by suppliers. If needed, powdery mildew species names were revised to reflect the current nomenclature.**

| Strain designation* | Host plant species | Host powdery mildew species | Place and year of isolation | GenBank accessions | | |
|---|---|---|---|---|---|---|
| | | | | ITS | *act1* | *eukNR* |
| ATCC 201056 | *Lycium barbarum* | *Arthrocladiella mougeotii* | Budapest, Hungary; 1990 | AF035780 | JN621873 | PQ838623 |
| A10a | *Lycium barbarum* | *Arthrocladiella mougeotii* | Budapest, Hungary; 2007 | HM124896 | PQ838577 | PQ838619 |
| A11a | *Lycium barbarum* | *Arthrocladiella mougeotii* | Budapest, Hungary; 2007 | HM124897 | PQ838579 | PQ838620 |
| A47b | *Lycium barbarum* | *Arthrocladiella mougeotii* | Budapest, Hungary; 2007 | HM124921 | PQ838580 | PQ838621 |
| A109a | *Lycium barbarum* | *Arthrocladiella mougeotii* | Budapest, Hungary; 2007 | HM124945 | PQ838581 | PQ838622 |
| CBS 132347 | *Vitis vinifera* | *Erysiphe necator* | Piacenza, Italy; 2009 | JN417714 | JN621822 | PQ838635 |
| CBS 132219 | *Vitis vinifera* | *Erysiphe necator* | Jesi, Italy; 2009 | JN417738 | JN621846 | PQ838631 |
| CBS 132220 | *Vitis vinifera* | *Erysiphe necator* | Jesi, Italy; 2009 | JN417739 | JN621847 | PQ838632 |
| Vitis79 | *Vitis vinifera* | *Erysiphe necator* | Portonovo, Italy; 2009 | JN417743 | JN621851 | PQ838643 |
| CBS 132224 | *Vitis vinifera* | *Erysiphe necator* | Eger, Hungary; 2009 | JN417752 | JN621860 | PQ838633 |
| CBS 132225 | *Vitis vinifera* | *Erysiphe necator* | Eger, Hungary; 2009 | JN417753 | JN621861 | PQ838634 |
| GYER | *Carpinus betulus* | *Erysiphe arcuata* | Budapest, Hungary; 2008 | HM124983 | MH879022 | MH879020 |
| DSM 2222 | *Cucumis* sp. | *Golovinomyces* sp. | Germany** | U82450 | JN621871 | PQ838636 |
| CBS 131.31 | *Helianthus tuberosus* | *Golovinomyces* sp. | USA; 1931 | AF035781 | PQ838587 | PQ838629 |
| CBS 133.32 | *Lonicera* sp. | *Erysiphe* sp. | USA; 1932 | HM124974 | PQ838588 | PQ838630 |
| CBS 129.79 | *Cucurbita pepo* | *Podosphaera xanthii* | Canada; 1975 | HQ108038 | PQ838585 | PQ838627 |
| CBS 130.79 | *Cucurbita pepo* | *Podosphaera xanthii* | Canada; 1975 | U82449 | PQ838586 | PQ838628 |
| RS1a | *Rosa* sp. | *Podosphaera pannosa* | Budapest, Hungary; 2007 | HM125010 | JN621896 | MW570719 |
| Ru1b | *Rudbeckia* sp. | *Golovinomyces* sp. | Salföld, Hungary; 2007 | HM125006 | PQ838590 | PQ838639 |
| RU2a | *Rudbeckia* sp. | *Golovinomyces* sp. | Salföld, Hungary; 2007 | HM125007 | PQ838591 | PQ838640 |
| Ru4b | *Rudbeckia* sp. | *Golovinomyces* sp. | Salföld, Hungary; 2007 | HM125008 | PQ838592 | PQ838641 |
| 263 | *Artemisia absinthium* | *Golovinomyces* sp. | Canada; 1974 | AF035782 | PQ838574 | PQ838615 |
| 3616Aa | *Plantago lanceolata* | *Podosphaera plantaginis* | Aland Archipelago, Finland; 2013 | KM066096 | PQ838576 | PQ838617 |
| 9031Aa | *Plantago lanceolata* | *Podosphaera plantaginis* | Aland Archipelago, Finland; 2013 | KM066093 | PQ838577 | PQ838618 |
| 2931Aa | *Plantago lanceolata* | *Podosphaera plantaginis* | Aland Archipelago, Finland; 2013 | KM066092 | PQ838575 | PQ838616 |
| HMLAC 05119 | *Youngia japonica* | Undetermined powdery mildew | China** | Extracted from the genome*** | Extracted from the genome*** | Extracted from the genome*** |
| BgrA | Undetermined grass | *Blumeria* sp. | Brno, Czechia; 2009 | PQ813604 | PQ838582 | PQ838624 |
| BgrB | Undetermined grass | *Blumeria* sp. | Brno, Czechia; 2009 | PQ813605 | PQ838583 | PQ838625 |
| BgrC | Undetermined grass | *Blumeria* sp. | Brno, Czechia; 2009 | PQ813606 | PQ838584 | PQ838626 |
| Trb | *Trifolium* sp. | *Erysiphe trifolii* | Budapest, Hungary; 2007 | PQ813608 | PQ838593 | PQ838642 |
| LS1a | *Lagerstroemia* sp. | *Erysiphe australiana* | Cestas, France; 2014 | PQ813607 | PQ838589 | PQ838638 |

*ATCC: American Type Culture Collection, Manassas, Virginia, USA (https://www.atcc.org/); CBS: Culture collection of the Westerdijk Fungal Biodiversity Institute, Utrecht, Netherlands (https://wi.knaw.nl/); DSM: Leibniz Institute DSMZ – German Collection of Microorganisms and Cell Cultures, Braunschweig, Germany (https://www.dsmz.de/). Strains without ATCC, CBS or DSM codes were not deposited in public culture collections.

**Missing data.

***Extracted from the genomic database (GenBank acc. no. VOSX00000000.1).

For ITS amplification, primers ITS1F [53] and ITS4 [54] were used as part of the PCR protocol described by Németh et al. [21]. The *act* fragment was amplified with primers Act-1 and Act-5ra [55] and the following PCR conditions: initial denaturation at 95ºC for 5 min followed by 38 cycles of 95ºC for 30 s, 60ºC for 1 min and 68ºC for 1 min. A final incubation of 68ºC for 5 min followed. The amplification of the *eukNR* fragment required development, as below.

To develop primers specific to the *eukNR* fragment in *Ampelomyces* spp., the target region was amplified first with a robust, general nested PCR method using primers niaD01F – niaD04R, followed by a second, nested amplification with the degenerate primers niaD15F – niaD12R and sequencing as described by Gorfer et al. [56]. This process was done with five genetically distinct *Ampelomyces* strains and the PCR products were sequenced and analysed. Based on the sequences obtained, primers niaD31F (CCGTCAGAAAGAGTAAAGGGTTT), niaD31F-alt (TCGTCCGGAAAAGCAAAG-GGTTT) and niaD32R (CAATACACTCCAGTACATGTCACG) were designed. Primers niadD31F and niaD32R worked well with most *Ampelomyces* strains, except BRIP 72097 that needed the primer combination niaD31F-alt – niaD32R to amplify its *eukNR* region. In all cases, PCR conditions were: 95ºC for 2.5 min followed by 38 cycles of 95ºC for 30 s, 60ºC for 20 s and 68ºC for 1 min followed by a final denaturation of 68ºC for 5 min.

## Sequence alignments and phylogenetic analyses

The datasets of each of the three target loci included the sequences of all the Australian isolates and 32 reference *Ampelomyces* strains isolated overseas and selected based on previous studies (Table 2). To obtain the ITS, *act* and *eukNR* sequences from HMLAC 05119, one of the two *Ampelomyces* strains with a published genome assembly [45], a local BLAST database was built using its assembled genome in Geneious Prime v.2025.0.3 (http://www.geneious.com) and the gene sequences generated in this study were queried against the local database. Seqtk (v.1.3-r106) was used to process the gene sequences.

After compiling the datasets, first each of the three loci was analysed separately. Sequences were aligned with Clustal W (v.1.83) [57]. Sequences were trimmed to the length of the shortest sequence to eliminate missing data and the Clustal alignment was repeated. In the new alignments, Clustal X (v.2.1) was used to visualise the alignments and change their format to a nexus format file. These alignments were used to infer genealogic trees using MrBayes (v.3.2.7a) with Bayesian inference to construct the trees [58] with settings nst = 6 and rates = invgamma (nucleotide substitution model SYM + I + gamma) for all loci. The Markov chain Monte Carlo analysis ran until the probability value decreased to under 0.01 with a sampling frequency of ten and a burn-in of 25% of samples. Trees were visualised with FigTree (v.1.4.4) (https://github.com/rambaut/figtree/). The figures were curated with Inkscape (https://inkscape.org). The three sets of sequences, i.e., ITS, *act* and *eukNR*, were concatenated and used to construct a phylogenetic network using the software SplitsTree4 and the NeighborNet algorithm [59], which allows for intergenic recombination. The concatenated alignment is available as S1 Data.

## DNA extraction, whole genome sequencing (WGS) and RNAseq

High molecular weight (HMW) DNA was extracted from lyophilised mycelia of *Ampelomyces* strains BRIP 72097 and BRIP 72102 using a chloroform/isoamyl alcohol protocol followed by isopropanol precipitation. Briefly, 100 mg of lyophilised mycelia was flash-frozen in liquid nitrogen and ground with stainless steel beads using a FastPrep-24 instrument. The ground material was lysed in pre-warmed lysis buffer containing potassium metabisulfite, Tris-HCl, EDTA, NaCl and CTAB, along with Sarcosyl, and incubated at 65ºC for 30 minutes. DNA was extracted using chloroform/isoamyl alcohol and precipitated with isopropanol. RNase treatment was performed at 37ºC for 2 hours, and the DNA was cleaned up using AMPure XP beads. Final purification was carried out using the Qiagen Genomic-tip 20/G kit, and DNA quality and quantity were assessed using a Qubit flurometer, Denovix spectrophotometer, and agarose gel electrophoresis.

Long-read sequencing was performed using Oxford Nanopore Technology (ONT). Libraries were prepared using the Genomic DNA by Ligation kit (SQK-LSK109) and sequenced on a MinION FLO-MIN106 R9.4.1 flow cell for 39 hours.

Read quality was assessed using NanoPlot. Short-read sequencing was conducted on the Illumina MiSeq platform. Libraries were prepared using the Illumina DNA Prep kit and Nextera DNA CD Indexes and sequenced using a 600-cycle paired-end V3 reagent kit. Read quality was evaluated using FastQC. Read preparation, genome assembly and annotation were performed as previously described by Huth et al. [24]. Briefly, raw reads were screened and filtered for bacterial contamination using Kraken v.2.1.1 [60]. Adaptors were removed from Illumina reads using BBDuk [61] and from the MinION reads with Porechop v.0.2.4 [62]. The hybrid assembler MaSuRCA v.3.3.3 [63] was used and the program OcculterCut v.1.1 [43] scanned the genomes to determine their percent GC content distribution. The genome size estimation using raw sequence data was conducted using k-mer analysis (k: 31) with Illumina short reads using the Galaxy Server tools Meryl (genomic k-mer counter and sequence utility; Galaxy Version 1.3 + galaxy6) and GenomeScope (reference-free genome profiling; Galaxy Version 2.0.1 + galaxy0) [64].

Total RNA extraction from fresh fungal mycelia were conducted according to Huth et al. [24] and submitted to the Australian Genome Research Facility (Melbourne, Australia) for total mRNA sequencing. Briefly, the mycelia were flash frozen, ground in liquid nitrogen and extracted using an RNeasy Plant Mini Kit (Qiagen) according to the manufacturer's instructions. The final products were checked via agarose gel electrophoresis and quantified using a Qubit v.3.0 fluorometer (ThermoFisher Scientific, Australia). Transcriptome assembly was conducted using Trinity v.2.10.0 [65] and genome annotation using Maker v.2.3.31.9 [66] including a first round of RNA-evidence gene prediction. The resulting annotation was used to produce a hidden Markov model (HMM), which was further refined with a second round of SNAP [67] training for the final annotation. The completeness of the genome assembly was evaluated via Benchmarking Universal Single-Copy Orthologs (BUSCO) v.5.8.0 [68] with the *dothideomycetes_odb10* lineage dataset, which contains 3,786 single-copy ortholog genes. Genome completeness was also assessed using the predicted protein dataset and BUSCO ran in protein mode with the *dothideomycetes_odb10*.

## Results

### *Ampelomyces* spp. strains isolated in Australia

Twenty-one *Ampelomyces* spp. strains were isolated from eight host plants and eight powdery mildew species in southern Queensland, and were deposited as live cultures at the Plant Pathology Herbarium, Queensland (BRIP) (Table 1). Apart from BRIP 72107 that was included in a WGS project [24], no other *Ampelomyces* strains were reported from Australia prior to this study. Collection sites included roadside areas covered mostly with weeds (Fig 1A, B); parks and gardens with ornamental plants (Fig 1C); and also broadacre cropping systems as three strains, BRIP 72966, BRIP 72967, and BRIP 72968, were isolated from powdery mildew-infected mungbean leaves collected from an irrigated commercial paddock (Table 1).

Mycoparasitic tests confirmed that four strains that sporulated in culture, i.e., BRIP 72097, BRIP 72100, BRIP 72107 and BRIP 72110, and isolated from four different powdery mildew and plant species (Table 1), were all able to produce intracellular pycnidia in *E. vignae* on mungbean plantlets (Fig 2B). The ITS sequences determined in the re-isolated mycoparasites were identical to those of the strains used for inoculations; therefore, Koch's postulates were fulfilled with those four strains.

### ITS genealogy and phylogenetic network analysis

The ITS sequences of the 21 *Ampelomyces* strains isolated in Australia (Table 1) were analysed together with 32 reference *Ampelomyces* strains isolated overseas in previous studies (Table 2), resulting in a 497-character long alignment. The ITS genealogy revealed that these 53 strains clustered into seven MOTUs (Fig 3). MOTU numbers used in this paper followed a previous study [24]. MOTU 1 included the majority of the newly isolated Australian strains, 13 in total, and BRIP 72107, as well, with an already published genome [24], together with 16 reference strains isolated from diverse powdery mildew species in Europe, the USA and Middle East. MOTU 4 contained three Australian strains isolated from mungbean

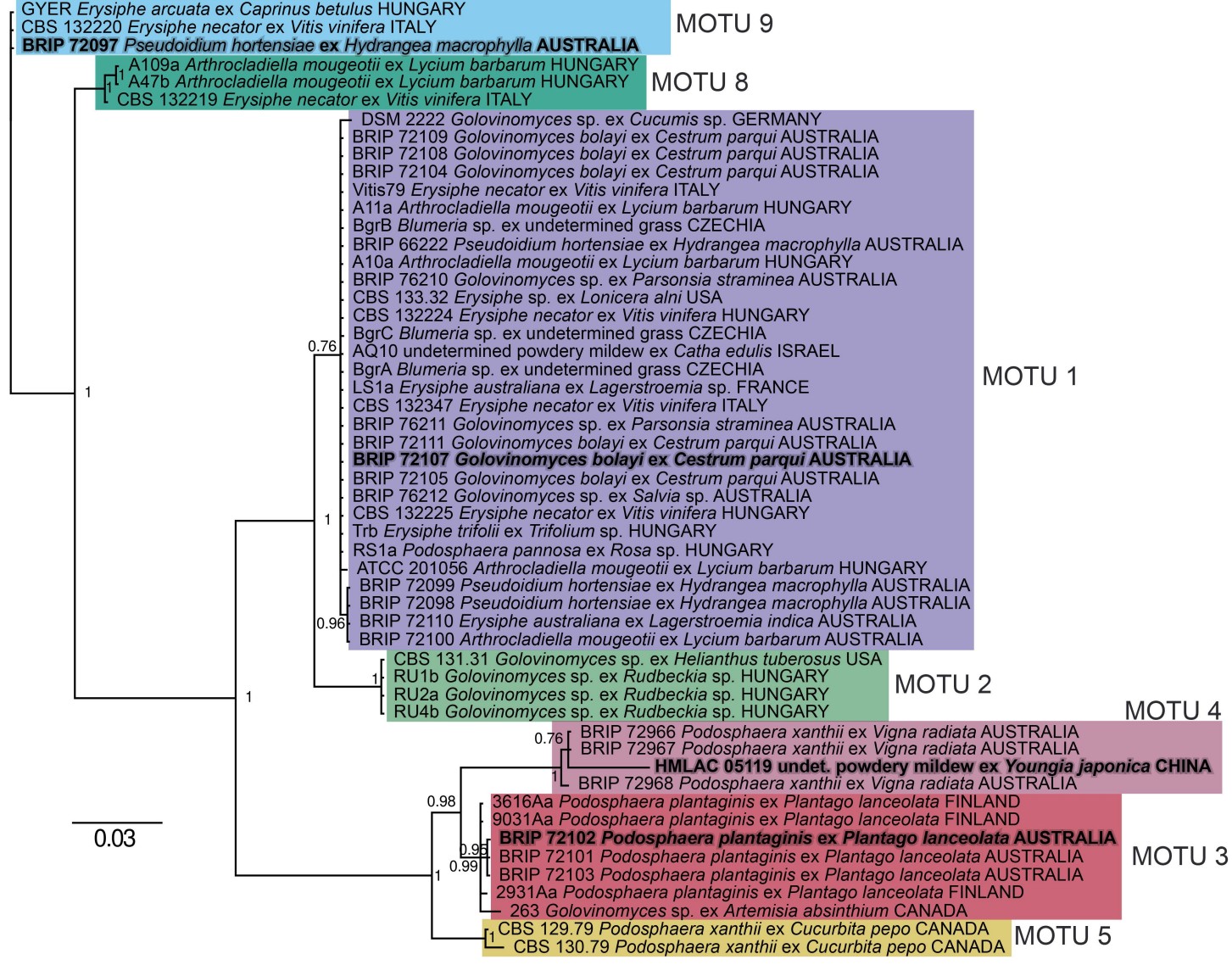

**Fig 3. Unrooted tree of the 21 Australian and 32 reference *Ampelomyces* strains based on nrDNA ITS sequence analyses.** Clustal W software was used for the ITS alignment and then Bayesian inference was used to infer the tree. Molecular taxonomic unit (MOTU) numbers follow a previous study (24). Each MOTU is highlighted with a different colour. The four *Ampelomyces* strains that have whole genome sequencing assemblies are indicated in bold and grey.

leaves infected with *P. xanthii* in a commercial paddock. The two Australian strains used for WGS, BRIP 72097 and BRIP 72102, were part of MOTU 9 and MOTU 3, respectively. Two more Australian strains, both isolated from *P. plantaginis* similar to BRIP 72102, were also included in MOTU 3.

MOTUs 2, 5, 6, 7 and 8 delimited in a previous study [24] did not include any strains from Australia. Some of the reference strains from overseas that represented these four MOTUs were available for the present study; therefore, these lineages were identified in this work, too ([Fig 3]). On the other hand, MOTUs 6 and 7 identified in the previous study [24] are missing from this work because none of their reference strains were available for *act* and *eukNR* sequencing that would have been needed for the multi-locus analysis.

As expected based on previous studies [17,18], the *act* sequence analysis revealed additional clades compared to the ITS genealogy; and some strains, including the commercial strain AQ10 and the Australian strain BRIP 72079, belonged to clades that were different from the MOTUs defined in the ITS analysis (S1 Fig). The *eukNR* genealogy provided a somewhat different grouping of the same 53 strains (S2 Fig). A multi-locus analysis was also performed on a dataset that included the *act*, *eukNR* and ITS sequences and was based on the NeighborNet algorithm using SplitsTree4. The concatenated alignment had a total length of 2,100 characters (*act*: 742 characters; *eukNR*: 861 characters; and ITS: 497 characters) with 1,717 identical (*act*: 647 characters; *eukNR*: 731 characters; and ITS: 339 characters) and 383 polymorphic (*act*: 95 characters; *eukNR*: 130 characters; and ITS: 158 characters) sites. The results of the phylogenetic network analysis are shown using SplitsTree4 (Fig 4). The clustering of the strains was mostly congruent with the ITS genealogy, i.e., the multi-locus work has also identified MOTUs 1, 2, 4, 5, 8 and 9 delimited by the ITS analysis. In addition, the multi-locus analysis split the ITS MOTU 3 into two groups. Three Australian strains, including BRIP 72102 selected for WGS, and strain 2931Aa isolated from the same powdery mildew and plant host in Finland, clustered together and were identified as a new group, MOTU 10. The other three strains from ITS MOTU 3 belonged to a closely related group designated as MOTU 3 in the multi-locus analysis (Fig 4).

## Assembly and annotation of two genomes

The Australian *Ampelomyces* strains BRIP 72097 and 72102 were selected for WGS in this work as these strains represent MOTUs 9 and 10, respectively. These two MOTUs are not closely related to each other and neither to MOTUs 1 and 4 that include each a strain with already available genomic information (Figs 3, 4).

*Ampelomyces* sp. strain BRIP 72097 was assembled into 22 scaffolds with a total assembly size of 33,451,943 bp and genome 'completeness' using Benchmarking Universal Single-Copy Orthologs (BUSCO) of 96.0% (Table 3). When compared to the predicted genome size of 35,397,803 bp using GenomeScope, genome completeness is estimated to be 94.5%. Of the total of 3,786 Dothideomycetes BUSCOs searched, BRIP 72097 included 3,634 complete and single-copy (95.9%), two complete and duplicated (0.1%), 14 fragmented (0.4%) and 138 missing (3.6%) BUSCOs. Based on a genome size of 33.45 Mb, and a total of 4.2 and 3.8 Gb of sequence data generated by MinION and Illumina MiSeq platforms, respectively, we estimated an approximate genome coverage of 240×. A combination of *ab initio* and evidence-based gene modelling with two additional rounds of gene predictions after training SNAP in the Maker pipeline resulted in 28,916 predicted exons within 10,417 genes, including 4,206 with 3′ untranslated regions (UTRs) and 4,553 with 5′ UTRs. Of the total of 3,786 Dothideomycetes BUSCOs searched using the predicted proteins, BRIP 72097 included 3,513 complete and single-copy (92.7%), four complete and duplicated (0.1%), 91 fragmented (2.4%) and 182 missing (4.8%) BUSCOs.

The other *Ampelomyces* sp. strain, BRIP 72102, was assembled into 38 scaffolds with a total assembly size of 37,322,005 bp and genome completeness of 95.9% (Table 3). When compared to the predicted genome size of 39,698,299 bp using GenomeScope, genome completeness is estimated to be 94.0%. Of the total of 3,786 Dothideomycetes BUSCOs searched, BRIP 72102 included 3,633 complete and single-copy (95.8%), five complete and duplicated (0.1%), 15 fragmented (0.4%) and 138 missing (3.7%) BUSCOs. Based on a genome size of 37.32 Mb, and a total of 3.8 and 14.2 Gb of sequence data generated by MinION and Illumina MiSeq platforms, respectively, we estimated an approximate genome coverage of 480×. A combination of *ab initio* and evidence-based gene modelling with two additional rounds of gene predictions after training SNAP in the Maker pipeline resulted in 29,360 predicted exons within 10,637 genes, including 3,974 with 3′ UTRs and 4,459 with 5′ UTRs. Of the total of 3,786 Dothideomycetes BUSCOs searched using the predicted proteins, BRIP 72102 included 3,530 complete and single-copy (93.2%), five complete and duplicated (0.1%), 79 fragmented (2.1%) and 172 missing (4.5%) BUSCOs.

Analysis of the assembled genomes for their distributions of AT and GC richness revealed their bipartite structure, consisting of gene-sparse AT-rich regions interspersed within gene-rich AT-balanced genomic regions (S3 Fig).

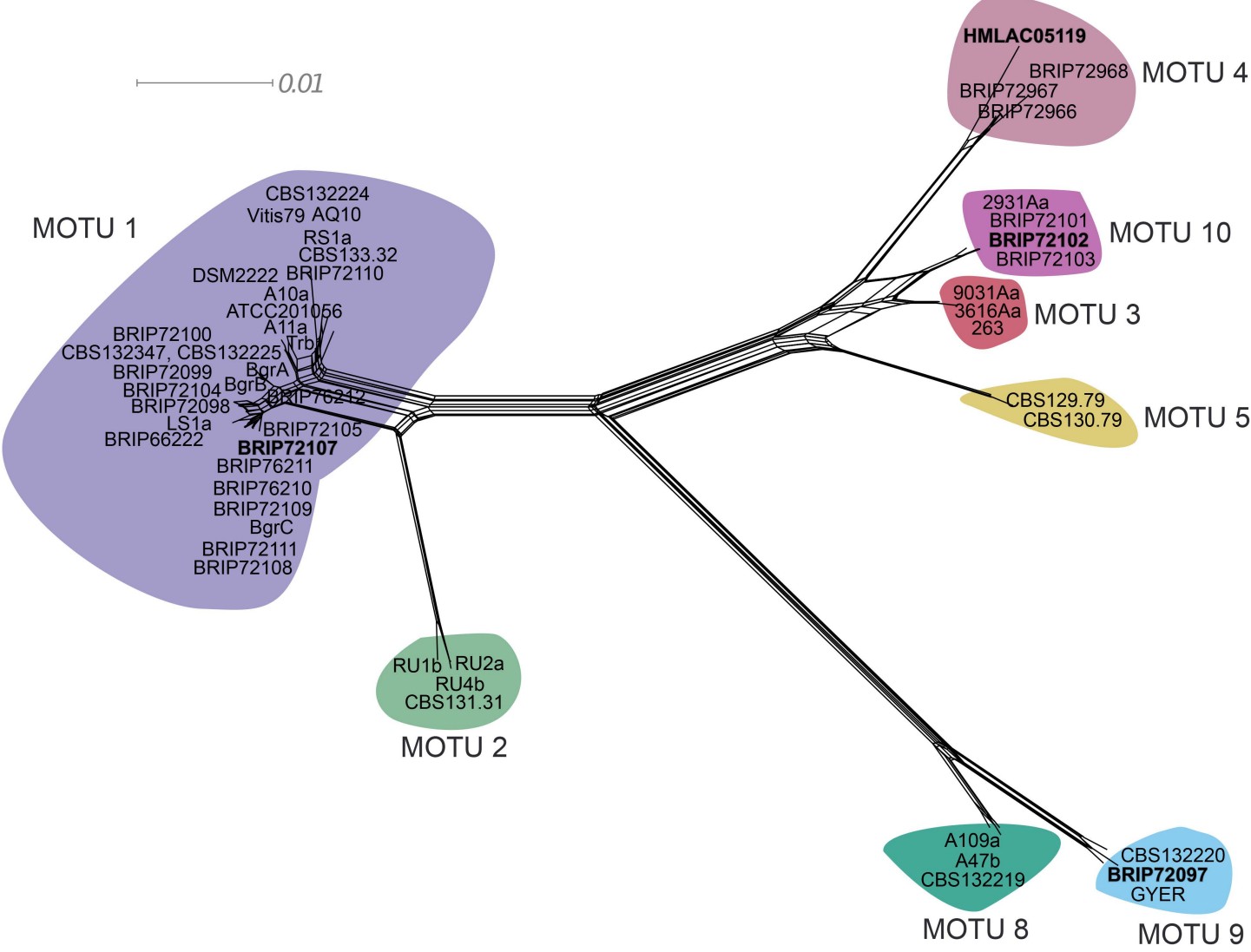

**Fig 4. A phylogenetic network based on the concatenated alignment of sequences of three loci (a fragment of the actin gene, a fragment of the nitrate reductase gene, and the nrDNA ITS region) of the 21 Australian and 32 reference *Ampelomyces* strains.** The network was generated using the NeighborNet algorithm in the software SplitsTree4. Molecular taxonomic unit (MOTU) numbers follow a previous study (24). Each MOTU is highlighted with a different colour. MOTU colour codes in this figure and Fig 3 are identical. The four *Ampelomyces* strains that have whole genome sequencing assemblies are indicated in bold and grey.

The percentages of AT-rich regions in the assembled genomes of BRIP 72097 and BRIP 72102 were 21% and 27%, respectively.

## Discussion

A recent hypothesis suggested that all powdery mildew species recorded in Australia so far were introduced inadvertently since 1788, the beginning of the European colonisation of the continent [50,69]. That particular year is considered as a sharp biogeographic landmark in the history of the Australian land vegetation [70], as it was the beginning of both deliberate and accidental human-assisted introductions of altogether over 28,000 plant species from overseas, including crops,

**Table 3. Genome statistics for the four *Ampelomyces* strains sequenced to date.**

| Strain[a] | Host plant species | Host powdery mildew species | Assembly size (Mb) | Cov[b] | No. of contigs | Contig N50 (bp) | No. of scaf-folds | Scaffold N50 (bp) | No. of Ns per Mb | GC content (%) | Genome comple-teness (%)[c] | NCBI accession number |
|---|---|---|---|---|---|---|---|---|---|---|---|---|
| BRIP 72097 | *Hydrangea macrophylla* | *Pseudoidium hortensiae* | 33.45 | 240 | 22 | 1,889,696 | 22 | 1,889,696 | 0 | 47.9 | 96.0 | JBBBEJ000000000 |
| BRIP 72102 | *Plantago lanceolata* | *Podosphaera plantaginis* | 37.32 | 480 | 38 | 3,841,740 | 38 | 3,841,740 | 0 | 46.2 | 95.9 | JBIFGS010000000 |
| BRIP 72107 | *Cestrum parqui* | *Golovino-myces bolayi* | 40.38 | 400 | 25 | 2,994,887 | 24 | 2,994,887 | 2 | 45.5 | 96.6 | JAGTXZ000000000 |
| HMLAC 05119 | *Youngia japonica* | Undeter-mined powdery mildew | 36.81 | 103 | 468 | 258,565 | 73 | 4,300,649 | 9,771 | 46.5 | 96.3 | VOSX00000000.1 |

[a]Strain BRIP 72107 was sequenced by Huth et al. (2021). HMLAC 05119 was obtained from the JGI Genome Portal (Haridas et al. 2020).

[b]Genome coverage

[c]Genome completeness for the two genomes generated in this work, BRIP 72097 and BRIP 72102, was determined based on benchmarking universal single-copy orthologs (BUSCOs) (Simao et al. 2015) against the dothideomycetes_odb10 database.

ornamentals, and pasture species [69–72]. It appears that powdery mildews were absent from Australia prior to 1788, and species of the *Erysiphaceae* were introduced accidentally to this continent in association with the massive introduction of their host plants [50,69]. If this hypothesis is true, all *Ampelomyces* mycoparasites must have been introduced to Australia since 1788, as well, together with their mycohosts, as the only known niche of these mycoparasites is inside powdery mildew colonies [8–10,40]. In Australia, *Ampelomyces* mycoparasites were first reported based on light microscopy observations of a number of powdery mildew species in Queensland in the 1960s [48]. However, only BRIP 72107 has been reported from Australia prior to this study [24].

This paper describes the isolation and characterisation of twenty new *Ampelomyces* strains. These were isolated in southern Queensland, in a relatively small geographical area, from only eight host plant species, all infected with different powdery mildew species representing four genera of the *Erysiphaceae*. Despite its limitations, the sampling revealed important information about *Ampelomyces* in Australia. First, the strains belonged to diverse MOTUs, four in total, that indicated the presence of multiple cryptic *Ampelomyces* species within a small geographic region. Second, phylogenetically different strains were isolated from the same powdery mildew species infecting the same host plant species; i.e., the strains coming from *P. hortensiae* belonged to two MOTUs. Also, one out of the three strains isolated from *P. xanthii*, in the same mungbean paddock, exhibited differences in all available marker regions (ITS, *act* and *eukNR*) to the two other strains, although the three strains clustered together in the ITS and the multi-locus analyses. Third, both the Australian and the reference strains belonging to MOTUs 1 and 9 were isolated from diverse powdery mildew species, representing different genera, while those in MOTU 4, and all but one strain in MOTU 3 appeared to be associated with a single mycohost species, *P. xanthii* and *P. plantaginis*, respectively. Fourth, all four Australian strains, and originating from four different powdery mildew species, included in the mycoparasitic tests were able to parasitize the mycohost species *E. vignae*, confirming their capacity to parasitize a species different from their original mycohosts. All these results are in agreement with the findings of previous studies on the phylogenetic diversity and mycoparasitic activity of *Ampelomyces* in relatively small geographic areas, such as Hungary [10,15,18], the Åland archipelago in Finland [12], Shandong, Sichuan and Shaanxi provinces in China [16], South Korea [17], Mie, Shiga and Tochigi Prefectures in Japan [21], and northern Italy [46].

Importantly, this study did not identify any MOTUs that had not been delimited in earlier studies based on strains isolated overseas. These results may indicate that *Ampelomyces* mycoparasites were introduced to Australia only very recently, together with their mycohosts.

One of the main goals of the phylogenetic analyses conducted in this study was to support the selection of more *Ampelomyces* strains for WGS. Strain BRIP 72107, sequenced earlier [24], belonged to MOTU 1; therefore, strains representing other MOTUs were prioritized for new WGS projects. The high-quality genome assemblies constructed for two strains representing ITS MOTUs 3 and 9 revealed their bipartite structure, i.e., presence of AT-rich, gene-sparse regions interspersed with GC-balanced, gene-rich regions, similar to BRIP 72107 [24]. The genome sizes of the three sequenced Australian *Ampelomyces* strains were markedly different, ranging from 33 to 40 Mb. This paper is also a 'genome announcement' [73] by providing two new high-quality genome assemblies and evidence-based annotations. The two new assemblies are important resources for future genomic analyses of mycoparasitic interactions to disentangle molecular mechanisms underlying mycoparasitism, possible new biocontrol applications, and towards the understanding of natural tritrophic relationships.

## Supporting information

**S1 Fig. Unrooted tree of the 21 Australian and 32 reference *Ampelomyces* strains based on actin gene (*act*) sequence analyses.** Clustal W software was used for the *act* alignment and then Bayesian inference was used to infer the tree. The four *Ampelomyces* strains that have whole genome sequencing assemblies are indicated in bold.
(PPTX)

**S2 Fig. Unrooted tree of the 21 Australian and 32 reference *Ampelomyces* strains based on eukaryotic nitrate reductase gene (*eukNR*) sequence analyses.** Clustal W software was used for the *eukNR* alignment and then Bayesian inference was used to infer the tree. The four *Ampelomyces* strains that have whole genome sequencing assemblies are indicated in bold.
(PPTX)

**S3 Fig. The GC content distribution of genome assemblies of *Ampelomyces* strains BRIP 72097 and BRIP 72102.** Vertical blue lines indicate the GC cut-off points selected by OcculterCut (43) to classify genome segments into distinct AT-rich and GC-balanced regions. The percent values on the left and right sides of the graphs indicate the percentage of the genome classified as AT-rich and GC-balanced, respectively.
(PPTX)

**S1 Data. Alignment of the concatenated *act* and *eukNR* and ITS sequences of 53 *Ampelomyces* strains analysed in this work.**
(NXS)

## Author contributions

**Conceptualization:** Levente Kiss.

**Data curation:** Lauren Goldspink, Alexandros G. Sotiropoulos, Márk Z. Németh, Niloofar Vaghefi, Levente Kiss.

**Formal analysis:** Lauren Goldspink, Alexandros G. Sotiropoulos, Niloofar Vaghefi.

**Funding acquisition:** Gavin J. Ash, Levente Kiss.

**Investigation:** Lauren Goldspink, Alexandros G. Sotiropoulos, Alexander Idnurm, John D. W. Dearnaley, Morwenna Boddington, Aftab Ahmad, Márk Z. Németh, Alexandra Pintye, Markus Gorfer, Hyeon-Dong Shin, Gábor M. Kovács, Niloofar Vaghefi, Levente Kiss.

**Methodology:** Lauren Goldspink, Alexandros G. Sotiropoulos, Alexander Idnurm, Markus Gorfer, Niloofar Vaghefi, Levente Kiss.

**Project administration:** Levente Kiss.

**Resources:** John D. W. Dearnaley, Morwenna Boddington, Márk Z. Németh, Alexandra Pintye, Markus Gorfer, Hyeon-Dong Shin, Gábor M. Kovács, Levente Kiss.

**Supervision:** Alexander Idnurm, Gavin J. Ash, Niloofar Vaghefi, Levente Kiss.

**Visualization:** Alexandros G. Sotiropoulos, Levente Kiss.

**Writing – original draft:** Lauren Goldspink, Alexandros G. Sotiropoulos, Márk Z. Németh, Niloofar Vaghefi, Levente Kiss.

**Writing – review & editing:** Lauren Goldspink, Alexandros G. Sotiropoulos, Alexander Idnurm, John D. W. Dearnaley, Alexandra Pintye, Markus Gorfer, Gábor M. Kovács, Niloofar Vaghefi, Levente Kiss.

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
