## [Decision Letter · Decision Letter 0]

28 May 2025

Dear Dr. Kiss,

Thank you for submitting your manuscript to PLOS ONE. After careful consideration, we feel that it has merit but does not fully meet PLOS ONE’s publication criteria as it currently stands. Therefore, we invite you to submit a revised version of the manuscript that addresses the points raised during the review process.

We look forward to receiving your revised manuscript.

Kind regards,

Kandasamy Ulaganathan

Academic Editor

PLOS ONE

 [This study was supported by Discovery Project DP210103869 of the Australian Research Council and the University of Southern Queensland, Australia.].

Additional Editor Comments (if provided):

Reviewers' comments:

Reviewer's Responses to Questions

**Comments to the Author**

1. Is the manuscript technically sound, and do the data support the conclusions?

Reviewer #1: Yes

Reviewer #2: Partly

2. Has the statistical analysis been performed appropriately and rigorously?

Reviewer #1: Yes

Reviewer #2: N/A

3. Have the authors made all data underlying the findings in their manuscript fully available?

Reviewer #1: Yes

Reviewer #2: No

4. Is the manuscript presented in an intelligible fashion and written in standard English?

Reviewer #1: Yes

Reviewer #2: Yes

Reviewer #1: The article entitled “Multi-locus phylogenetic network analysis of Ampelomyces mycoparasites isolated from diverse powdery mildews in Australia and the generation of two de novo genome assemblies” is well witten. It provides detailed analysis on Ampelomyces from diverse powdery mildews in Australia. It is an excellent work that offers valuable contribution to the scientific community in this field. I recommend accepting the article after minor revisions. Here are the suggestions:

1. Abbreviations should be defined at their first appearance in the text. A few abbreviations are not explained such as SNAP and UTRs.

2. The value "4.3" listed as the Contig N50 (bp) for HMLAC 05119 in Table 3 appears to be incorrect. Please verify whether it should be "258,565," as reported in your previous study.

Other minor corrections:

Line 108: “infected with on Podosphaera xanthii” instead of “infected with onPodosphaera xanthii”.

Line 194: There is plants "Mungbean (Vigna radiata) plants cv. Jade" but "(Vigna radiata) cv. Jade-AU" in the figure legend. Make it consistent.

Line 378-379: Check if there need a period here: “BUSCO of 96.0 % (Table 3) Of the total”; check also Line 397-398.

Reviewer #2: This paper represents an important molecular contribution to the study of Ampelomyces mycoparasites. Genetic data has been generated for multiple Australian strains, improving understanding of relationships between strains and their potential origins. Two strains were selected for whole genome assembly and annotation, which appear to be of high quality.

Overall, the rationale and overarching narrative of the paper is clear. However, the impact and clarity of the manuscript has been substantially affected by a lack of some key details and analyses. As a consequence, in its current form, some of the stated conclusions are not supported by the data presented.

A. Essential/Recommended revisions

A1. The Introduction is generally clear and well-written. However, I felt that the genetics/genomics of Ampelomyces needs a bit more detail and clarification, given the focus of the paper. Firstly, it would be useful to be more explicit about the life cycle - the is reference to an asexual life cycle and a sexual morph, but it is not clear how these relate to each other. It is important to be clear about the ploidy, if known, as this has implications for sequencing and assembly. Furthermore, the karyotype should be discussed, even if just to acknowledge that it is unknown. (If this is the case, are there related fungi from which inferences can be made?) Lastly, the loci used for previous molecular phylogenies (L68-70) that define the MOTUs should be listed. Is this based on a few loci, or phylogenomics?

A2. The phylogeny of Figure 3 is interesting but needs a little additional work. In addition to the network from concatenated data (Figure 4), I would like to see the trees of the other two marker genes too for consistency, possibly as supplementary data. From Fig 3, it appears that a lot of the sequences are identical, and there are a very small number of differences within each MOTU. Is the same true for the other genes? I am therefore not convinced (L368) that “The analysis has indicated possible recombination amongst some MOTUs”, as it looks like this is based on very little signal. Furthermore, it is not strictly true that (L366) "The clustering of the strains was congruent with the ITS genealogy, i.e., the multi-locus analysis revealed the same clusters (MOTUs) as the analysis of the ITS sequences" nor that (L344) “The ITS genealogy revealed that these 53 strains clustered into eight MOTUs (Fig 3).” MOTU 3 and MOTU 10 form a single cluster in Fig 3 and cannot be split into separate clades. Figure 3 also needs to be rooted (is it possible to date any of the splits?) or drawn more explicitly as an unrooted tree like Fig 4. (At present it appears to be unrooted but is drawn as if MOTU 9 is an outgroup to the rest, which is not correct.)

L292: "These alignments were used to create genealogic trees … " This should be “infer phylogenetic trees”. Please provide the alignments as supplementary data. How was the substitution model selected? Ideally, a complementary method (NJ or ML) would also be used to check for robustness to assumptions.

A3. Analysis/validation of the two new genomes is incomplete. Assembly sizes should be compared to predicted genome sizes from raw data (e.g. kmer-based like GenomeScope and/or depth-based like DepthSizer) and experimental genome size if known (see A1). Without this, the statement in L461 cannot be supported: “The genome sizes of the three sequenced Australian Ampelomyces strains were markedly different, ranging from 33 to 40 Mb." (This is an observed difference in assembly size, not genome size.)

The kmer completeness, QV and ploidy should also be reported (e.g. Merqury and GenomeScope). It would be good to have a sense as to whether these assemblies are approaching chromosome-length. (What is the karyotype?) Telomere prediction might help with this. Similarly, it would be good to know if 10k genes is lot for this kind of species, or what one would expect. BUSCO completeness should be calculated for the predicted proteome and/or transcriptome, and annotations should also be provided as supplementary data. Basic repeat annotation should also be performed.

A4. The authors have done a lot of work selecting and assembling the two new Australian strains of Ampelomyces, increasing the number of MOTUs with an assembly to four. However, the paper terminates prematurely, without any analysis or comparison of the four genomes. Whilst detailed phylogenomic analysis is probably beyond the scope of this paper, it should be fairly easy to test whether shared BUSCO single copy orthologues give a phylogenetic signal that is consistent with (a) the three marker genes used for defining MOTUs, and (b) each other. This should cover around a third of the genes (if the annotation is complete) and thus give a strong indication whether there seems to be recombination or horizontal gene transfer between these lineages. The contiguity of the assemblies appears to be quite high, so it would also be good to perform some basic synteny analysis - again, the BUSCO genes can be used for this.

A5. L406: “Analysis of the assembled genomes for their distributions of AT and GC richness revealed their bipartite structure, consisting of gene-sparse AT-rich regions interspersed within gene-rich AT-balanced genomic regions. The percentages of AT-rich regions in the assembledgenomes of BRIP 72097 and BRIP 72102 were 21% and 27%, respectively." This is incomplete and needs more detail and visualisation. How are these regions defined and distributed? Something like a Circos plot might be useful.

A6. The assemblies do not appear to be publicly available on Genbank. The authors need to provide details of the BioProject, BioSample and assembly accession numbers.

B. Minor revisions

B1. L74-76. "Until this is done, the use of Ampelomyces spp. is recommended when referring to phylogenetically diverse strains within the genus." It would be good here to highlight the lack of knowledge about recombination (L93-94), and how robust the species concept is likely to be in Ampelomyces.

B2. L108, typo: "onPodosphaera".

B3. L115: "The highest quality assembly and annotation…" It would be informative to summarise how good this assembly is. Is it chromosome-level, for example?

B4. L182: "were identified as two distinct Golovinomyces species" This should be marked in Table 1.

B5. Table 2: "Extracted from the genome***" Please provide the genome scaffold accession numbers and positions.

B6. L303: “High molecular weight (HMW) DNA extraction and long-read sequencing of BRIP 72097 andBRIP 72102 was performed using Oxford Nanopore Technology (ONT) and short-read sequencing performed using the Illumina MiSeq platform with read preparation, genome assembly and annotation as previously described by Huth et al. (24)." The methods here are not clear. Extraction and sequencing should be separated and the essentials of the technologies used (e.g. ONT kit & flowcell, Illumina library & cycles) and assembly methods provided even if more details are given in the citation.

B7. L320: “The completeness of the genome assembly was evaluated via Benchmarking Universal Single-Copy Orthologs (BUSCO) v.1.2 (66).” State the lineage used and number of genes.

B8. Table 3. The contig N50 for HMLAC05119 cannot be 4.3 bp.

**Do you want your identity to be public for this peer review?** For information about this choice, including consent withdrawal, please see our Privacy Policy

Reviewer #1: **Yes: ** Yuan-Min Shen

Reviewer #2: **Yes: ** Richard Edwards

---

## [Author Response · Author response to Decision Letter 1]

14 Aug 2025

Professor Kandasamy Ulaganathan

Centre for Plant Molecular Biology, Osmania University, Hyderabad, India

Academic Editor, PLoS ONE

Dear Professor Ulaganathan,

Thank you for your comments on our work and judging it as suitable for publication in PLoS ONE pending satisfactory revision.

Please find below our responses and reactions to the comments from Reviewers #1 and #2. Line (L) numbers in their comments refer to our original submission while line numbers in our responses refer to the file uploaded as ‘Revised Manuscript with Track Changes’.

We would like to sincerely thank both Reviewers for their time spent on our manuscript and all their thoughtful comments and suggestions that have been very helpful during the revision of this work.

The manuscript was fully revised in line with their comments, and we think that it has been improved considerably during revision. All changes made in our original submission are shown with tracked changes in the file uploaded as ‘Revised Manuscript with Track Changes’.

The file uploaded as ‘Manuscript’ was generated by accepting all tracked changes shown in ‘Revised Manuscript with Track Changes’.

As detailed in our responses, we think that some of the queries listed by Reviewer #2 referred to further analyses that were beyond the scope of this study. The overarching goal of this work is to:

a) reveal the genetic diversity of Ampelomyces strains isolated in Australia in order to support strain selection for whole-genome sequencing (WGS) projects; and

b) release the newly assembled genomes of two phylogenetically different strains.

This was detailed in the last paragraph of the Introduction.

We are currently performing comprehensive phylogenomic analyses of a number of Ampelomyces genomes, including those two that are reported in this manuscript. The results will be included in another paper; therefore, we cannot fulfil some of the queries listed by Reviewer #2. This was explained in our response to Reviewer #2.

Thank you for considering this revision. We do hope that the revised version is suitable for publication in PLoS ONE.

Kind regards,

Prof Levente Kiss

corresponding author for this submission

Response to Reviewer #1

>COMMENT: The article entitled “Multi-locus phylogenetic network analysis of Ampelomyces mycoparasites isolated from diverse powdery mildews in Australia and the generation of two de novo genome assemblies” is well witten. It provides detailed analysis on Ampelomyces from diverse powdery mildews in Australia. It is an excellent work that offers valuable contribution to the scientific community in this field. I recommend accepting the article after minor revisions.

RESPONSE: Thank you for your positive feedback.

>COMMENT: 1. Abbreviations should be defined at their first appearance in the text. A few abbreviations are not explained such as SNAP and UTRs.

REPLY and REACTION:

• SNAP is not an abbreviation – it is the name of a web server. We added the reference to the first mention of SNAP in the text (L352; Korf 2004, reference #67 – other references were re-numbered as needed following this change).

• UTR (untranslated region) was written in full at first mention (L434).

• No other unexplained abbreviations were found in the text.

>COMMENT: 2. The value "4.3" listed as the Contig N50 (bp) for HMLAC 05119 in Table 3 appears to be incorrect. Please verify whether it should be "258,565," as reported in your previous study.

REPLY and REACTION: Corrected in Table 3.

>COMMENT: Line 108: “infected with on Podosphaera xanthii” instead of “infected with onPodosphaera xanthii”.

RESPONSE AND REACTION: ‘on’ deleted.

>COMMENT: Line 194: There is plants "Mungbean (Vigna radiata) plants cv. Jade" but "(Vigna radiata) cv. Jade-AU" in the figure legend. Make it consistent.

RESPONSE AND REACTION: Agreed, the correct name of the variety is ‘Jade-AU’. Corrected throughout the manuscript and Table 1.

>COMMENT: Line 378-379: Check if there need a period here: “BUSCO of 96.0 % (Table 3) Of the total”; check also Line 397-398.

RESPONSE AND REACTION: Corrected.

Response to Reviewer #2

>COMMENT: This paper represents an important molecular contribution to the study of Ampelomyces mycoparasites. Genetic data has been generated for multiple Australian strains, improving understanding of relationships between strains and their potential origins. Two strains were selected for whole genome assembly and annotation, which appear to be of high quality.

RESPONSE: Thank you for this positive comment.

>COMMENT: Overall, the rationale and overarching narrative of the paper is clear. However, the impact and clarity of the manuscript has been substantially affected by a lack of some key details and analyses. As a consequence, in its current form, some of the stated conclusions are not supported by the data presented.

RESPONSE: The manuscript was revised in line with this general comment. The revised version should now not contain any conclusions that are not supported by the data provided.

A. Essential/Recommended revisions

>COMMENT: A1. The Introduction is generally clear and well-written. However, I felt that the genetics/genomics of Ampelomyces needs a bit more detail and clarification, given the focus of the paper. Firstly, it would be useful to be more explicit about the life cycle - the is reference to an asexual life cycle and a sexual morph, but it is not clear how these relate to each other.

RESPONSE AND REACTION:

• The asexual life cycle of Ampelomyces is well understood and described in detail in the papers referenced in L77-92. References #9 and #10, cited in this paragraph, include detailed figures of the life cycle. During revision, we added #9 one more time, to the last sentence of this paragraph, to provide an even more comprehensive reference list of the asexual life cycle.

• There is a recent observation of a fruiting body reported as the sexual morph of Ampelomyces. We cited this paper (reference #27) without comments in the original submission. In our view, reference #27 is somewhat unclear regarding the identity of that fungal structure. The relevant part was revised (L94-97) to highlight that this is a single observation; and the sexual morph of Ampelomyces has never been reported elsewhere.

>COMMENT: It is important to be clear about the ploidy, if known, as this has implications for sequencing and assembly.

RESPONSE AND REACTION: A sentence was added (L118-119) to state that the hyphae and conidia of Ampelomyces are haploid, similar to most other ascomycetes.

>COMMENT: Furthermore, the karyotype should be discussed, even if just to acknowledge that it is unknown. (If this is the case, are there related fungi from which inferences can be made?)

RESPONSE AND REACTION:

• A sentence was added to state that the chromosome numbers of Ampelomyces spp. have not been revealed yet (L119). Another sentence was added to L139-140 to specify that none of the two genome assemblies available before our study (i.e., those of HMLAC 05119 and BRIP 72107) were near-chromosome level assemblies.

• For your information, we would like to add the following:

We have recently sequenced the genomes of a few more phylogenetically different strains using long-reads with PacBio HiFi; and assembled their genomes at chromosome level. Our unpublished assemblies were supported by a Hi-C study, as well. These results are part of a comprehensive genomic study that will be deposited soon in bioRxiv, and submitted for publication in a relevant journal.

• Chromosome numbers have been reported in some of the relatives of Ampelomyces but may not be useful references for reasons revealed by our unpublished genomic study.

>COMMENT: Lastly, the loci used for previous molecular phylogenies (L68-70) that define the MOTUs should be listed. Is this based on a few loci, or phylogenomics?

RESPONSE AND REACTION: Previous phylogenies were based on ITS and actin gene (act) sequences. This information was added to L69-70.

>COMMENT: A2. The phylogeny of Figure 3 is interesting but needs a little additional work. In addition to the network from concatenated data (Figure 4), I would like to see the trees of the other two marker genes too for consistency, possibly as supplementary data.

RESPONSE AND REACTION: Single-locus trees based on act and eukNR sequences, respectively, were added to the manuscript as Supplementary Figures 1 and 2. The results of the act and the eukNR analyses were briefly mentioned in L396-400.

>COMMENT: From Fig 3, it appears that a lot of the sequences are identical, and there are a very small number of differences within each MOTU. Is the same true for the other genes? I am therefore not convinced (L368) that “The analysis has indicated possible recombination amongst some MOTUs”, as it looks like this is based on very little signal.

RESPONSE AND REACTION: We agree that our analysis did not reveal convincing signals of recombination. Therefore, all comments regarding recombination were deleted during revision from the Abstract, Results, and Discussion.

For your information, our currently ongoing phylogenomics analyses, mentioned above, are considering recombination signals, in addition to a number of other analyses.

>COMMENT: Furthermore, it is not strictly true that (L366) "The clustering of the strains was congruent with the ITS genealogy, i.e., the multi-locus analysis revealed the same clusters (MOTUs) as the analysis of the ITS sequences" nor that (L344) “The ITS genealogy revealed that these 53 strains clustered into eight MOTUs (Fig 3).” MOTU 3 and MOTU 10 form a single cluster in Fig 3 and cannot be split into separate clades.

RESPONSE AND REACTION: Agreed: our conclusions in the original submission were quite superficial in this respect. To rectify these issues, the following revisions were done:

• ITS MOTUs 3 and 10 were combined into MOTU 3 in Fig. 3. The text was modified accordingly: in L381, we now state that the ITS analysis has identified seven MOTUs, not eight. Other changes were made in L387 and L389 to replace ITS MOTU 10 with 3.

• In L407, we specified that “The clustering of the strains was mostly congruent with the ITS genealogy” and introduced MOTU 10 there (L408-413) as a result of the multi-locus analysis.

>COMMENT: Figure 3 also needs to be rooted (is it possible to date any of the splits?) or drawn more explicitly as an unrooted tree like Fig 4. (At present it appears to be unrooted but is drawn as if MOTU 9 is an outgroup to the rest, which is not correct.)

RESPONSE AND REACTION: We prefer to keep the ITS (and also the act and the eukNR) single-locus trees as unrooted due to our difficulties to identify the same outgroup for these three analyses. The revised captions of Fig. 3 (the ITS tree) and Suppl. Figs 1 and 2 (the act and the eukNR trees, respectively) highlight that these are all unrooted. This is also indicated by their revised drawings.

>COMMENT: L292: "These alignments were used to create genealogic trees … " This should be “infer phylogenetic trees”.

RESPONSE AND REACTION: Corrected in the text and also in the caption of Fig. 3.

>COMMENT: Please provide the alignments as supplementary data. How was the substitution model selected? Ideally, a complementary method (NJ or ML) would also be used to check for robustness to assumptions.

RESPONSE AND REACTION: The multi-locus alignment is now provided as Suppl. Data 1. Substitution models were compared in MEGA.

>COMMENT: A3. Analysis/validation of the two new genomes is incomplete. Assembly sizes should be compared to predicted genome sizes from raw data (e.g. kmer-based like GenomeScope and/or depth-based like DepthSizer) and experimental genome size if known (see A1). Without this, the statement in L461 cannot be supported: “The genome sizes of the three sequenced Australian Ampelomyces strains were markedly different, ranging from 33 to 40 Mb." (This is an observed difference in assembly size, not genome size.)

RESPONSE AND REACTION: Genome sizes were estimated using GenomeScope as described in the revised Mat & Meth (L337-342, with reference #64 added – the references from this point were re-numbered as needed). Genome completeness statistics based on GenomeScope were added to the Results (L425-426 and L449-451). The original statements regarding genome sizes still hold true.

>COMMENT: The kmer completeness, QV and ploidy should also be reported (e.g. Merqury and GenomeScope). It would be good to have a sense as to whether these assemblies are approaching chromosome-length. (What is the karyotype?) Telomere prediction might help with this. Similarly, it would be good to know if 10k genes is lot for this kind of species, or what one would expect. BUSCO completeness should be calculated for the predicted proteome and/or transcriptome, and annotations should also be provided as supplementary data. Basic repeat annotation should also be performed.

RESPONSE AND REACTION: BUSCO completeness was repeated using the predicted proteome and results added to the manuscript (L434-437 and L459-461). Annotations for the two strains were added as Supplementary Data 2 and 3 (mentioned in L437-438 and L461-462). The assemblies are still not near-chromosome level, and as mentioned above, the karyotype is unknown in Ampelomyces strains.

As mentioned above, our follow-up study generated chromosome-level assemblies and should be published soon. This manuscript is needed as a foundation work for our more comprehensive genomic analyses.

>COMMENT: A4. The authors have done a lot of work selecting and assembling the two new Australian strains of Ampelomyces, increasing the number of MOTUs with an assembly to four. However, the paper terminates prematurely, without any analysis or comparison of the four genomes. Whilst detailed phylogenomic analysis is probably beyond the scope of this paper, it should be fairly easy to test whether shared BUSCO single copy orthologues give a phylogenetic signal that is consistent with (a) the three marker genes used for defining MOTUs, and (b) each other. This should cover around a third of the genes (if the annotation is complete) and thus give a strong indication whether there seems to be recombination or horizontal gene transfer between these lineages. The contiguity of the assemblies appears to be quite high, so it would also be good to perform some basic synteny analysis - again, the BUSCO genes can be used for this.

RESPONSE AND REACTION: At the end of the Discussion, we have edited the comment “This paper serves as a ‘genome announcement” (L521-522). Therefore, we hope the paper does not ‘terminate prematurely’ but rather serves to set a platform for more sophisticated investigations using additional Ampelomyces genomes in comparison to close relatives. In our view, the requests to analyse single copy orthologues (i.e., to provide a genome-wide phylogeny) and carry out other analyses, as well, are beyond the scope of this paper.

For your information, we are currently performing such analyses that include the two new assemblies reported in this manuscript. The results will be included in a future paper, mentioned above.

>COMMENT: A5. L406: “Analysis of the assembled genomes for their distributions of AT and GC richness revealed their bipartite structure, consisting of gene-sparse AT-rich regions interspersed within gene-rich AT-balanced genomic regions. The percentages of AT-rich regions in the assembled genomes of BRIP 72097 and BRIP 72102 were 21% and 27%, respectively." This is incomplete and needs more detail and visualisation. How are these regions defined and distributed? Something like a Circos plot might be useful.

RESPONSE AND REACTION: We now added to the manuscript that we used OcculterCut to estimate GC content distribution (L341-342). OcculterCut applies a sliding-window approach to calculate GC% across the genome and uses a Gaussian mixture model to separate sequences into high-GC and low-GC components. We have provided the OcculterCut results as Supplementary Figure 3 (mentioned in L461-462).

>COMMENT: A6. The assemblies do not appear to be publicly available on Genbank. The authors need to provide details of the BioProject, BioSample and assembly accession numbers.

RESPONSE AND REACTION:

---

## [Decision Letter · Decision Letter 1]

7 Sep 2025

Dear Dr. Kiss,

Thank you for submitting your manuscript to PLOS ONE. After careful consideration, we feel that it has merit but does not fully meet PLOS ONE’s publication criteria as it currently stands. Therefore, we invite you to submit a revised version of the manuscript that addresses the points raised during the review process.

We look forward to receiving your revised manuscript.

Kind regards,

Kandasamy Ulaganathan

Academic Editor

PLOS ONE

Journal Requirements:

Reviewers' comments:

Reviewer's Responses to Questions

**Comments to the Author**

Reviewer #1: All comments have been addressed

Reviewer #2: All comments have been addressed

2. Is the manuscript technically sound, and do the data support the conclusions?

Reviewer #1: Yes

Reviewer #2: Yes

3. Has the statistical analysis been performed appropriately and rigorously?

Reviewer #1: Yes

Reviewer #2: Yes

4. Have the authors made all data underlying the findings in their manuscript fully available?

Reviewer #1: Yes

Reviewer #2: Yes

5. Is the manuscript presented in an intelligible fashion and written in standard English?

Reviewer #1: Yes

Reviewer #2: Yes

Reviewer #1: The article "Multi-locus phylogenetic network analysis of Ampelomyces mycoparasites isolated from diverse powdery mildews in Australia and the generation of two de novo genome assemblies" provide important detailed analysis on Ampelomyces from diverse powdery mildews in Australia. It is an great work that offers valuable contribution to the scientific community in this field. Minor revision is suggested since I agree with reviewer 2 that "Figure 3 also needs to be rooted or drawn more explicitly as an unrooted tree like Fig 4."

Reviewer #2: The authors have made a substantial effort to address the previous comments. I think the paper is clearer as a result. I only have a couple of minor residual comments/questions about the BUSCO analysis.

1. Did the authors really use BUSCO v1.2? I did not notice this the first time. BUSCO is now up to v6 and the more recent versions are much more consistent, so should be used preferentially. It does not matter too much for this paper, but the authors mention ongoing work, so they really should switch to at least v5 for that.

2. The proteome analysis uses a different orthdb dataset to the genome analysis. Please use the same for both. It is hard to judge the actual quality of the annotation if the stats relate to a different dataset.

**Do you want your identity to be public for this peer review?** For information about this choice, including consent withdrawal, please see our Privacy Policy

Reviewer #1: **Yes: ** Shen, Yuan-Min

Reviewer #2: No

---

## [Author Response · Author response to Decision Letter 2]

21 Oct 2025

Professor Kandasamy Ulaganathan

Centre for Plant Molecular Biology, Osmania University, Hyderabad, India

Academic Editor, PLoS ONE

Dear Professor Ulaganathan,

Thank you for your comments on our work and judging it as suitable for publication in PLoS ONE pending a 2nd round of revision.

Please find below our responses and reactions to the comments from Reviewers #1 and #2. Line (L) numbers in our responses refer to the file uploaded as ‘2nd Revision of Manuscript with Tracked Changes’.

In addition to minor changes done in line with the 2nd Reviewer’s comments (as detailed below), we have also done a few cosmetic changes in the manuscript – all shown with tracked changes.

We would like to sincerely thank both Reviewers for their time spent on our manuscript.

The file uploaded as ‘Manuscript’ was generated by accepting all tracked changes shown in ‘2nd Revision of Manuscript with Tracked Changes’.

Thank you for considering this 2nd revision. We do hope that the current version is suitable for publication in PLoS ONE.

Best regards,

Prof Levente Kiss

corresponding author for this submission

Response to Reviewer #1

>COMMENT:

The article "Multi-locus phylogenetic network analysis of Ampelomyces mycoparasites isolated from diverse powdery mildews in Australia and the generation of two de novo genome assemblies" provide important detailed analysis on Ampelomyces from diverse powdery mildews in Australia. It is an great work that offers valuable contribution to the scientific community in this field. Minor revision is suggested since I agree with reviewer 2 that "Figure 3 also needs to be rooted or drawn more explicitly as an unrooted tree like Fig 4."

RESPONSE: Thank you for your kind words. Concerning the three unrooted trees (Figure 3 – ITS tree; and Suppl. Figs. 1 and 2 – the act and the eukNR trees, respectively), these have to remain unrooted because of the difficulty of finding the same outgroup for all these three loci. This was explained in our previous response to Reviewer #2.

To highlight the unrooted nature of these three trees, the captions of all the three figures start with ‘Unrooted tree of …’.

We have experimented with replacing these three figures with network-like and circular trees. However, one of the values of these trees is to display all the details (designations, mycohosts, host plants, and countries of origin) of the mycoparasitic strains included in the analysis, and this can only be achieved in the current format of Figure 3 and Suppl. Figs. 1 and 2. The phylogenetic network (Figure 4) could not display all the information that is available on the unrooted trees.

This is the rationale for presenting these three figures in their current format.

Response to Reviewer #2

>COMMENT:

The authors have made a substantial effort to address the previous comments. I think the paper is clearer as a result. I only have a couple of minor residual comments/questions about the BUSCO analysis.

RESPONSE: Thank you for your positive feedback.

>COMMENT:

1. Did the authors really use BUSCO v1.2? I did not notice this the first time. BUSCO is now up to v6 and the more recent versions are much more consistent, so should be used preferentially. It does not matter too much for this paper, but the authors mention ongoing work, so they really should switch to at least v5 for that.

RESPONSE AND REACTION: Thank you for spotting this typo. We used BUSCO v5.8.0 and corrected the typo in L343.

>COMMENT:

2. The proteome analysis uses a different orthdb dataset to the genome analysis. Please use the same for both. It is hard to judge the actual quality of the annotation if the stats relate to a different dataset.

RESPONSE AND REACTION: Thank you for spotting this issue. The proteome BUSCO was run again with dothideomycetes_odb10 and the results were updated in the manuscript (L422-425 and L444-446).

---

## [Decision Letter · Decision Letter 2]

17 Nov 2025

Multi-locus phylogenetic network analysis of Ampelomyces mycoparasites isolated from diverse powdery mildews in Australia and the generation of two de novo genome assemblies

PONE-D-25-16889R2

Dear Dr. Kiss,

We’re pleased to inform you that your manuscript has been judged scientifically suitable for publication and will be formally accepted for publication once it meets all outstanding technical requirements.

Kind regards,

Kandasamy Ulaganathan

Academic Editor

PLOS ONE

Additional Editor Comments (optional):

Reviewers' comments:

Reviewer's Responses to Questions

**Comments to the Author**

Reviewer #1: All comments have been addressed

Reviewer #2: All comments have been addressed

2. Is the manuscript technically sound, and do the data support the conclusions?

Reviewer #1: Yes

Reviewer #2: Yes

3. Has the statistical analysis been performed appropriately and rigorously?

Reviewer #1: Yes

Reviewer #2: Yes

4. Have the authors made all data underlying the findings in their manuscript fully available?

Reviewer #1: Yes

Reviewer #2: Yes

5. Is the manuscript presented in an intelligible fashion and written in standard English?

Reviewer #1: Yes

Reviewer #2: Yes

Reviewer #1: The revision of the manuscript "Multi-locus phylogenetic network analysis of Ampelomyces mycoparasites isolated

from diverse powdery mildews in Australia and the generation of two de novo genome assemblies" is acceptable.

Reviewer #2: The authors have addressed all the comments. I would still present the unrooted tree slightly differently but it is their paper, not mine.

**Do you want your identity to be public for this peer review?** For information about this choice, including consent withdrawal, please see our Privacy Policy

Reviewer #1: **Yes: ** Yuan-Min Shen

Reviewer #2: **Yes: ** Richard J. Edwards

---

## [Editor Report · Acceptance letter]

PONE-D-25-16889R2

PLOS ONE

Dear Dr. Kiss,

I'm pleased to inform you that your manuscript has been deemed suitable for publication in PLOS ONE. Congratulations! Your manuscript is now being handed over to our production team.

Kind regards,

on behalf of

Dr. Kandasamy Ulaganathan

Academic Editor

PLOS ONE